# The Phenotype of Bone Turnover in Patients with Fragility Hip Fracture: Experience in a Fracture Liaison Service Population

**DOI:** 10.3390/ijerph19127362

**Published:** 2022-06-15

**Authors:** Carla Caffarelli, Nicola Mondanelli, Eduardo Crainz, Stefano Giannotti, Bruno Frediani, Stefano Gonnelli

**Affiliations:** 1Department of Medicine, Surgery and Neuroscience, Section of Internal Medicine, University of Siena, 53100 Siena, Italy; carlacaffarelli@yahoo.it; 2Department of Medicine, Surgery and Neurosciences, Section of Orthopedics and Traumatology, University of Siena, 53100 Siena, Italy; nicola.mondanelli@unisi.it (N.M.); stefano.giannotti@unisi.it (S.G.); 3Department of Orthopaedics and Traumatology, Section of Orthopedics and Traumatology, University Hospital of Siena, 53100 Siena, Italy; e.crainz@ao-siena.toscana.it; 4Department of Medicine, Surgery and Neurosciences, Rheumatology Unit, University of Siena, 53100 Siena, Italy; bruno.frediani@unisi.it

**Keywords:** hip fracture, Fracture Liaison Service, vitamin D, PTH, βCTX, Charlson Comorbidity Index, previous fragility fracture

## Abstract

Background: Hip fragility fractures are becoming one of the main health care problems in countries with an aging population. This study aimed to evaluate the clinical characteristics and the usefulness of bone turnover markers in patients with a hip fracture. Methods: In a cohort of 363 patients (84.1 ± 9.2 years) with hip fractures we measured 25-hydroxyvitamin D (25OHD), bone alkaline phosphatase, type I collagen β carboxy telopeptide (βCTX), and parathyroid hormone (PTH). We recorded patients’ Charlson Comorbidity Index (CCI) and previous history of fragility fractures. Results: Vitamin D and PTH levels were inversely correlated (r = −024; *p* < 0.001). The prevalence of 25OHD deficiency was 57.8%, the PTH levels greater than 65 pg/mL was in 47.0 %, and in those who had βCTX values the upper limit was 61.8%. Moreover, 62% of patients with a fragility hip fracture had a history of a previous fracture. The 25OHD serum levels were inversely associated with CCI and a previous fragility fracture. On the contrary, PTH and βCTX serum levels showed a positive significant correlation with CCI and previous fragility fractures. Conclusion: This study confirmed the usefulness of a bone turnover markers assessment, along with the comorbidities and history of previous fragility fractures in order to better identify the risk of hip fracture.

## 1. Introduction

Severe osteoporosis is considered one of the most common diseases world-wide, resulting in millions of fragility fractures every year; 25% of men and 50% of women over the age of 50 years will experience an osteoporosis-related fracture [1]. Fragility fractures in the elderly represent a serious event, with a significant impact on patients’ quality of life and considerable repercussions in terms of public health [1,2]. Fragility hip fractures are an emerging health problem, especially due to the progressive aging of the population. In fact, for the high morbidity and mortality, hip fractures have considerable health and economic consequences for patients and their families, and a burden for health care systems worldwide [3]. Another aspect is represented by the inadequate perception of the problem. It should instead be emphasized that, in Europe, the likelihood of running into one fragility fracture in the course of life is analogous to that of having a stroke [4]. In fact, the number of hospitalizations due to a fracture of the femur is also continuously increasing, as a reflection of the absolute number of fractures occurring in subjects ≥ 65 years of age and especially in those over the age of 75 (84.9% of cases), an age group in which both the prevalence of osteoporosis and the risk of falls markedly increase [5]. It has been suggested that patients with hip fragility fractures tend to be older, more osteoporotic, present hypovitaminosis D, and hyperparathyroidism [6]. Moreover, the majority of hip fracture patients present a history of previous fragility fractures. In fact, it has been reported that vertebral fracture increases the risk of subsequent hip fracture by at least twofold, whereas a history of forearm fracture increases the risk of future hip fracture by approximately 50% [7,8]. Moreover, previous studies have reported that lateral femoral fractures are more correlated with a severe degree of osteoporosis and are characterized by a more compromised recovery after hip fractures [9].

From the international literature, it is clear that despite the notable advances in surgical techniques, post-surgical management of the hip fracture is often inadequate, especially in the prevention of new fracture events. In fact, although the fragility fracture constitutes one of the major risk factors for the onset of further fractures, an appropriate diagnostic and therapeutic process is initiated only for a minority of patients [10]. In this context, a multidisciplinary approach that includes bone diseases competencies along with a well-coordinated team such as a Fracture Liaison Service (FLS) [11], may represent the best option for management. In fact, the Fracture Liaison Service is considered the most effective model of care for the prevention of recurrent fractures thanks to the adoption of coordinated and intensive strategies that guarantee adherence to treatment [10,12].

The aim of our study was: (1) to evaluate the characteristics of patients with hip fractures admitted to our hospital in the three years following the implementation of FLS and (2) to determine the usefulness of bone turnover markers in these patients.

## 2. Materials and Methods

### 2.1. Population

We studied a cohort of 363 patients (289 female and 74 male) 60 years or older, hospitalized because of a proximal native or low-impact femur fracture, referred to the Orthopedics Clinics at the University Hospital of Siena (Italy), and evaluated by the FSL team between January 2019 and December 2021. The hip fractures were classified by experienced clinicians on preoperative radiographs and surgical reports as medial fractures (or cervical or intracapsular) and lateral fractures (or trochanteric or extracapsular). The pathological hip fractures due to primary or metastatic bone cancer, multiple myeloma, Paget’s disease of bone, or primary hyperparathyroidism were excluded. The patients with chronic renal failure were also excluded. 

All patients underwent a clinical examination and a detailed history including demographics data, type of fracture, comorbidity profile, and Charlson Comorbidity Index (CCI) score at time of surgery [13]. CCI score evaluates patients’ comorbidity data, namely, the presence or absence of the specific comorbidity as previously diagnosed and recorded on their past medical history [13]. The comorbidities included cardiovascular, pulmonary, neurological, endocrine, renal, gastrointestinal, and malignant conditions. Each comorbidity is assigned a point weight age ranging from 1 to 6, and a summation of the scores will give rise to the overall CCI score [14]. Moreover, information was collected during FLS team visit on history of previous fragility fractures. All surveys included details of fracture location, including spine, hip, wrist, clavicle, upper arm/shoulder, rib, pelvis, ankle, upper leg, and lower leg. Besides, the occurrence of fractures was assessed by way chart review in the Carestream database, by reviewing progress notes, and by radiological evidence of vertebral fracture. Magnetic resonance imaging (MRI) and computed tomography (CT) reports were also reviewed.

### 2.2. Laboratory Evaluation

In all subjects, during the first five days of hospitalization after fracture occurrence, in the morning after an overnight fast, fasting venous blood samples were drawn in order to evaluate serum calcium (Ca), phosphate (P), creatinine (Cr), albumin, parathyroid hormone (PTH), 25-hydroxyvitamin D (25OHD), 1,25-dihydroxyvitamin D (1,25(OH)2D), bone alkaline phosphatase (B-ALP), and type I collagen β carboxy telopeptide (βCTX). Serum PTH was assessed by an immunoradiometric assay (Total Intact PTH Antibodies Lab. Inc.; Santee, CA, USA) and the intra- and inter-assay coefficients of variation were 3.6 and 4.9%, respectively. Serum 25OHD was determined by a chemiluminescence immunoassay (LIAISON 25OHD Total Assay, DiaSorin Inc., Stillwater, MN, USA). In our institution the intra- and inter-assay coefficients of variation were 6.8% and 9.2%, respectively. According with recent literature, we considered values below 12 ng/mL (30 nmol/L) deficient and to be associated with an increased risk of rickets/osteomalacia [15]. Moreover, levels of 25(OH)D between 12 and 20 ng/mL are considered to be in the insufficient range and levels between 20 ng/mL and 50 ng/mL (50–125 nmol/L) appear to be sufficient for the general healthy population [15,16]. Serum1,25(OH)2D was assessed by chemiluminescence immunoassay (LIAISON XL 1,25-Dihydroxyvitamin D, DiaSorin Inc., Stillwater, MN, USA). In our institution the intra- and inter-assay coefficients of variation were 4.1% and 5.3%, respectively. Serum B-ALP was measured by a chemiluminescence immunoassay method (LIAISON BAP Ostase, DiaSorin Inc., Stillwater, MN, USA). In our institution the intra-and inter-assay coefficients of variation for B-ALP were 4.2% and 7.9%, respectively. Serum βCTX was evaluated by an enzyme-linked immunoassay method (Immunodiagnostic Systems, Boldon, UK); in our institution the intra- and inter-assay coefficients of variation were 2.5% and 4.0%, respectively. Healthy controls were recruited from a sub-group (aged 70 years or more) of the elderly men and women living in the area of Siena (Italy), who had been participating in a larger epidemiological study [17]. 

### 2.3. Statistical Analysis

All values were expressed as mean ± SD. The Kolmogorov–Smirnov test was used to verify the normality of the distribution of the outcome variables. Clinical data and initial values of the variables measured in the study groups were compared using Student’s *t*-test and Mann–Whitney U-test as appropriate. Categorical variables were compared by chi-square test or Fisher’s exact test, as appropriate. The associations between different parameters were tested by either Pearson’s correlation or Spearman’s correlation as appropriate. 

An informed written consent was obtained from all participants, and the study was approved by the Institutional Review Board of Siena University Hospital. 

All tests were performed using the SPSS statistical package for Windows version 16.0 (SPSS Inc., Chicago, IL, USA). 

## 3. Results

The clinical and biochemical characteristics of the study population are reported in Table 1. Female subjects were slightly older and had serum levels of creatinine slightly lower with respect to male patients. In particular, our data showed reduced vitamin D serum level values with a simultaneous increase in serum PTH and βCTX levels values both in females and males. Moreover, there were no significant differences between the two groups in all biochemical parameters. 

The anthropometric and biochemical characteristics of hip fracture patients and healthy controls are reported in Table 2. As expected, hip fracture patients presented significantly lower 25OHD serum levels and significantly higher levels of PTH and β-CTX with respect to controls. 

Figure 1 shows the mean values of the Charlson score in female and male patients with a hip fragility fracture. It is evident that the CCI was significantly lower in female patients with respect to males (*p* < 0.01). 

Vitamin D and PTH levels were inversely correlated (r = −024; *p* < 0.001). The distribution of the serum levels of Vitamin D and PTH in the 363 patients with hip fractures are shown in Figure 2.

In particular, the overall prevalence of vitamin D deficiency, defined as 25OHD levels equal or less than 12 ng/mL (30 nmol/L) was 57.8%; insufficient range defined 25OHD as levels between 20 ng/mL and 50 ng/mL (50–125 nmol/L) as 30.2% and only 12.0 % of patients had levels of 25OHD indicative of sufficient vitamin D serum levels (20 ng/mL and 50 ng/mL (50–125 nmol/L). The percentage of PTH levels greater than 65 pg/mL was observed in 47.0 % of patients with a hip fragility fracture. In addition, the percentage of patients who had βCTX values had an upper limit of 61.8%. 

Moreover, 62% of patients with a fragility hip fracture had an history of previous fracture (Figure 3A); The distribution of fracture sites is displayed in Figure 3B. The most frequently reported sites were the vertebrae (*n* = 131), femur (*n* = 64), wrist (*n* = 58), and tibia/fibula (*n* = 57). 

Table 3 presents the age and gender-adjusted partial correlations of biochemical parameters with CCI and previous fragility fractures in patients with fragility hip fractures. The 25OHD serum levels were inversely associated with CCI and previous fragility fractures, without reaching statistical significance. On the contrary, PTH and βCTX serum levels showed a positive significant correlation with CCI and previous fragility fractures. No significant associations between B-ALP with CCI and previous fractures were observed. 

One hundred forty-seven patients had a medial fragility fracture, and 216 patients had a lateral hip fracture; the distribution of different hip fracture types is reported in Figure 4. 

There were no significant differences in 25OHD and B-ALP serum levels, instead PTH and βCTX serum levels were increased in the patients with lateral fragility fractures without reaching a significant difference (PTH = 87.8 ± 61.1 vs. 77.8 ± 47.5; βCTX = 1.104 ± 0.638 vs. 0.898 ± 0.560, respectively).

Figure 5 shows the percentage of patients with a “medial” or “lateral” hip fracture on the basis of the presence of previous fragility fractures. It is evident that the prevalence of previous fragility fractures was higher in patients with a lateral hip fracture than in those with a medial fracture (64.4% vs. 53.4 %, respectively, *p* < 0.05). 

## 4. Discussion

With the aging of the population and the high prevalence of osteoporosis, the incidence of hip fractures is predicted to increase progressively with an economical burden difficult to sustain [2]. Therefore, in order to plan preventive strategies, there is a growing interest in obtaining better improved risk and protective factors of hip fractures and define new care pathways for the integrated management of patients with hip fractures. 

The main finding of this study, carried out on a cohort of elderly patients with hip fractures, was that more than 60% of the patients had a history of previous fragility fractures and a deficiency in vitamin D with a marked increase in both PTH and βCTX serum levels. 

In agreement with other studies, in our patients, the prevalence of Vitamin D deficiency was particularly high [18,19,20,21,22]. In fact, using a cut-off of 25OHD of 12 ng/mL (30 nmol/L), vitamin D deficiency was present in 57.8% of patients, and only 12% presented with vitamin D serum levels in the normal range. No differences in vitamin D levels were observed between females and males and between the different types of hip fractures (lateral vs. medial). 

The vitamin D deficiency in our study population could be explained by the fact that only a minimal percentage of our patients were taking any type of vitamin D supplementation at the hip fracture time. However, the inadequate supplementation of vitamin D in Italian patients suffering from a hip fracture has been reported in several studies [20,23].

As expected, considering the marked hypovitaminosis D, about half of the patients presented PTH values higher than 65 pg/mL. However, the high rate of hyperparathyroidism in patients with a hip fracture is in agreement with the majority of previous studies [18,20,21,22,24]. Moreover, the marked increase in PTH has been reported to be independently associated with poor outcomes being associated with a high risk of complication, high mortality rates, and an increased risk of institutionalization [19,20,21,25]. In fact, it is well known that elevated PTH levels exert catabolic effects on bone and may be implicated in the genesis of bone fragility and hip fractures [26,27]. In particular, hyperparathyroidism is able to increase bone loss, especially at the cortical level, which is one of the major mechanisms involved in the hip fragility fracture [21]. However, a study by Seitz et al. reported that in the biopsies of the patients with a femoral neck fracture, the histomorphometric analysis revealed not only a decreased bone volume and trabecular thickness, but also a significant increase in both osteoid indices and the heterogeneity of mineralization [24]. Moreover, this latter study, reported that patients with a femoral neck fracture and serum 25OHD levels below 12 μg/L displayed significantly thinner trabecular bone [24]. Obviously, the values of PTH as well as hypovitaminosis D could be influenced by aging and renal insufficiency. 

This is one of the first studies evaluating βCTX in a large population of patients with a recent hip fracture. Another important finding of the study is that 60% of the subjects have high βCTX values and that βCTX values were significantly correlated with PTH values. Our data are in agreement with recent articles that have shown that the increase in serum βCTX values is related to bone fragility and can predict an increased risk of mortality [28,29]. The importance of the assessment of serum βCTX as a possible risk factor for hip fracture is supported by the fact that βCTX, similar to PTH, is significantly related to the CCI and the presence of previous fragility fractures [25,29]. On this basis, βCTX could be considered as a possible indicator of reduced bone strength at the femoral level [29].

As previously reported [30,31,32], in our study more than sixty percent of patients had a history of previous fragility fractures. In agreement with previous studies, we have found that the prevalence of previous fragility fractures was significantly higher in patients with a lateral than in those with a medial hip fracture [33,34]. In particular, Mautalen et al. reported that previous vertebral fractures were twice as frequent in patients with a lateral hip fracture with respect to those with a medial fracture [34]. The higher prevalence of fragility fractures in patients with a lateral femoral fracture could be explained by the fact that the trochanteric region is characterized by a high percentage of trabecular bone.

This study emphasizes that the major determinants of fracture risk should be taken into account in the management of patients with a hip fracture. This type of approach can only take place in the context of an integrated patient management as it occurs in an FLS. Therefore, our results underline the importance and effectiveness of adopting an FLS model for the clinical management. 

Our study has some limitations. First, the time and type of surgery, the type of anesthesia, and other factors that may influence results were not analyzed. Secondly, the lack of data on patients’ nutritional status and dietary calcium intake. Thirdly, the cross-sectional design does not prove causal inference. Nevertheless, our study presents several strengths. First, the fact that is a large and homogeneous sample size of elderly patients with a hip fracture. Secondly, we evaluated a complete panel of bone turnover parameters in all patients. 

## 5. Conclusions

In conclusion, our study confirmed the usefulness of the evaluation of the biochemical parameters along with comorbidity and history of previous fragility fracture in order to better identify the risk of hip fracture. Moreover, after a hip fracture the FLS can be considered the most appropriate organizational approach for the management of patients who suffer a hip fracture and prevent a secondary fracture. This is made possible only by patient education, osteoporosis treatment prescription, falls prevention, and long-term adherence improvement; all these points represent the mission of the FLS model.

## Figures and Tables

**Figure 1 ijerph-19-07362-f001:**
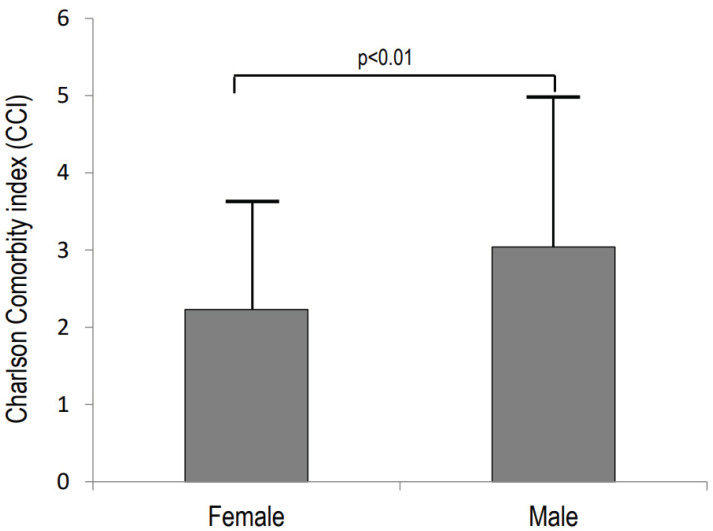
Average Charlson Comorbidity Index (CCI) values by gender in patients with hip fragility fracture.

**Figure 2 ijerph-19-07362-f002:**
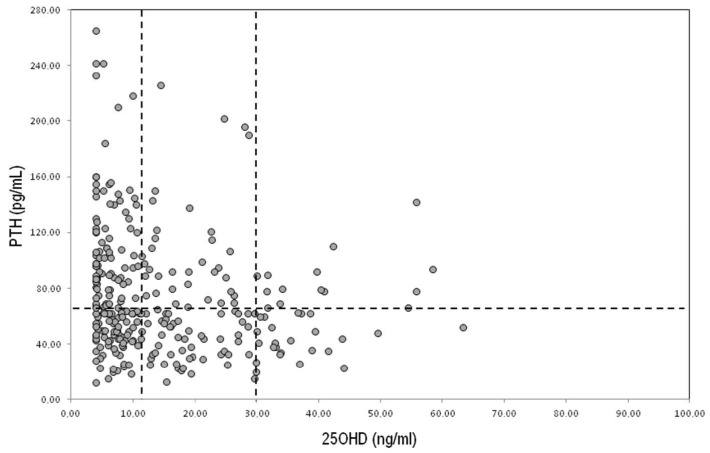
Parathyroid and 25OH vitamin D serum levels.

**Figure 3 ijerph-19-07362-f003:**
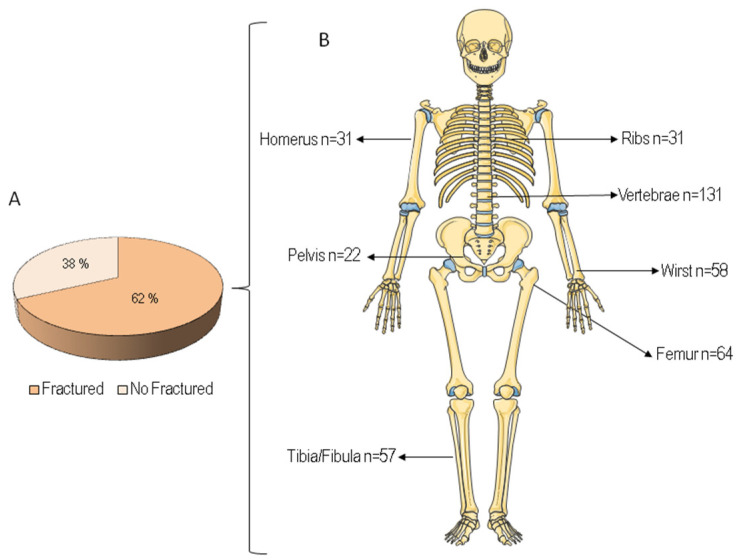
Previous fragility fractures in patients with hip fracture (**A**) and what the most common fracture sites were (**B**).

**Figure 4 ijerph-19-07362-f004:**
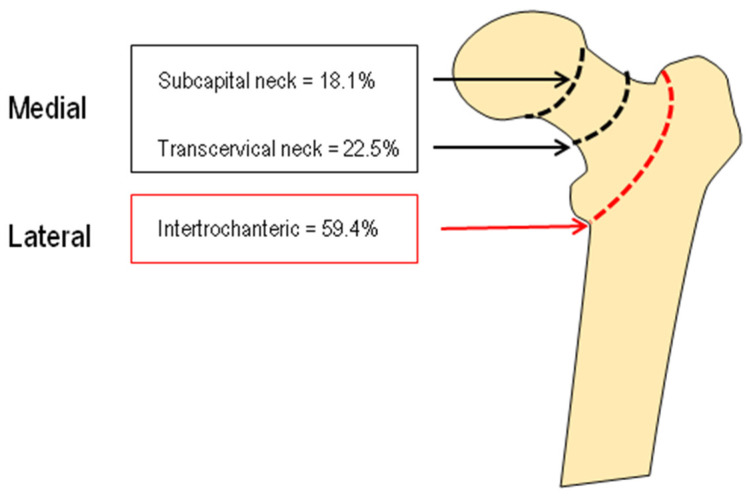
The types of hip fractures in the study population.

**Figure 5 ijerph-19-07362-f005:**
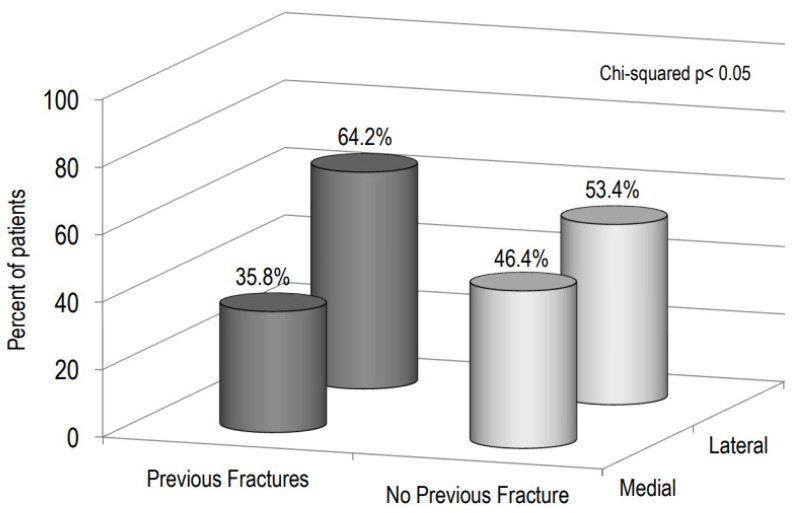
Percentage of patients with “medial” or “lateral” hip fracture on the basis of the presence of previous fragility fractures.

**Table 1 ijerph-19-07362-t001:** Demographic and biochemical characteristics of the study population.

	Female (*n* = 289)	Male (*n* = 74)
Age (years)	84.5 ± 9.2	82.4 ± 9.5
BMI (Kg/m^2^)	24.1 ± 4.9	24.6 ± 4.0
Creatinine (mg/dL)	0.91 ± 0.32	1.08 ± 0.46 *
Calcium (mg/dL)	8.24 ± 0.58	8.24 ± 0.51
Phosphate (mg/dL)	3.00 ± 0.79	2.90 ± 0.92
Albumin (g/dL)	3.15 ± 0.31	3.08 ± 0.30
ALP (UI/L)	85.61 ± 44.86	75.84 ± 28.66
25OHD (ng/mL)	15.45 ± 14.85	12.86 ± 9.10
1,25(OH)D2 (pg/mL)	36.68 ± 23.77	31.39 ± 22.30
PTH (pg/mL)	78.61 ± 57.01	75.02 ± 47.56
B-ALP (µg/L)	12.11 ± 7.52	9.15 ± 4.31
β-CTX (ng/L)	1.062 ± 1.018	0.988 ± 0.675

* *p* < 0.05 female vs. men.

**Table 2 ijerph-19-07362-t002:** Biochemical characteristics of the patients with hip fracture and the healthy controls.

	Hip Fracture (*n* = 363)	Controls (*n* = 194)
Sex (F/M)	289/74	140/54
Age (years)	84.0 ± 9.2	72.8 ± 2.6 **
BMI (Kg/m^2^)	24.1 ± 4.9	25.9 ± 2.9 *
Creatinine (mg/dL)	0.92 ± 0.32	0.95 ± 0.20 *
Calcium (mg/dL)	8.24 ± 0.56	9.24 ± 0.52 **
Phosphate (mg/dL)	2.98 ± 0.82	3.31 ± 0.60 **
ALP (UI/L)	83.62 ± 42.19	75.84 ± 28.66
25OHD (ng/mL)	14.90 ± 13.88	24.7 ± 9.10 **
PTH (pg/mL)	77.8 ± 55.01	23.99 ± 13.60 **
B-ALP (µg/L)	11.6 ± 7.10	11.80 ± 5.43
β-CTX (ng/L)	1.250 ± 0.500	0.616 ± 0.296 **

* *p* < 0.05; ** *p* < 0.001 Hip fracture vs. controls.

**Table 3 ijerph-19-07362-t003:** Age- and gender-adjusted partial correlations of Charlson Comorbidity Index and the presence of Previous Fragility Fracture with bone turnover markers, 25OHD, 1.25OHD, and PTH serum levels.

	Charlson Comorbidity Index	Previous Fragility Fracture
25OHD (ng/mL)	−0.042	−0.115
1.25(OH)D2 (pg/mL)	−0.066	−0.090
PTH (pg/mL)	0.178 **	0.271 *
B-ALP (µg/L)	0.079	−0.022
β-CTX (ng/L)	0.177 **	0.389 **

* *p* < 0.05; ** *p* < 0.01.

## Data Availability

Not applicable.

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
