# Peer review of "The Phenotype of Bone Turnover in Patients with Fragility Hip Fracture: Experience in a Fracture Liaison Service Population"

_ijerph, 2022, doi:10.3390/ijerph19127362_

Round 1
Reviewer 1 Report
A more extensive description/explanation should be added in the text regarding two step points:
- medial vs lateral fracture location in the casistics
- correlation of the focused parameter βCTX with fracture location and prognosis
Author Response
According to the suggestion of the Reviewer we added a sentence and a new Reference in the introduction “section” to emphasize the importance of the type of femoral fracture (lateral or medial). Moreover in the “results” section we added biochemical results on the basis of the fracture type.

Reviewer 2 Report
The topic is interesting. The study is well designed and the paper well organized.
I have a few suggestions to improve the quality of the paper.
Any difference between lateral and medial fractures?Please provide markers levels differentiated by fracture type
Please provide a comparison of used markers with a comparable population with no femur fractures. If not possible, please discuss with data from the Literature
The paper must be checked by a native speaker
Author Response
According to the suggestion of the Reviewer we added in the “results” section biochemical results on the basis of the fracture type.

Reviewer 3 Report
This manuscript is well written and the authors analyzed a large cohort of patients.
The study is interesting from an epidemiological point of view; though it lacks of originality and novelty.
I would add a graph to describe the study population in terms of types of fractures.
Author Response
According to the suggestion of the Reviewer we added Figure 4 that describe the study population in terms of types of fractures

Reviewer 4 Report
Thank you for the opportunity to review. I have the following suggestions:
Page 1 line 14 change to countries and aging, delete to before define
Page 1 line 17 and before parathyroid
Page 1 line 20 sentence needs to be restructured
Page 1 line 22 ICC should that be CCI
Page 1 line 32 delete in fact
Page 1 line 37 need to restructure sentence
Page 1 line 41 change have to having
Pahe 1 line 41 42 restructure sentence
Page 2 line 46 delete present
Page 2 line 50 change around to approximately
Page 2 line 62 delete twofold
Page 2 line 63 to determine the usefulness of bone turnover
Page 2 line 83 change to malignant
Page 3 line 103 change recent paper to recent literature
Page 3 line 131 change show to showed
Page 5 line 157 drop the in front of femur and wrist
Page 6 lines 165, 167, 168 change ICC to CCI
Page 7 line 187 change funding to finding
Page 7 line 186 change better improved
Page 7 line 201 Italian
Page 8 line 241 and line 245 change Firstly to First
Author Response
According to the suggestion of the Reviewer we made the changes.

Round 2
Reviewer 1 Report
Minor text editing is auspicable.
Reviewer 2 Report
The Authors made great efforts in the attempt to ameliorate their paper. It now merits publication